# Research Advances in the Role of the Tropomyosin Family in Cancer

**DOI:** 10.3390/ijms241713295

**Published:** 2023-08-27

**Authors:** Yucheng Meng, Ke Huang, Mingxuan Shi, Yifei Huo, Liang Han, Bin Liu, Yi Li

**Affiliations:** 1Key Laboratory of Dental Maxillofacial Reconstruction and Biological Intelligence Manufacturing, School of Stomatology, Lanzhou University, Lanzhou 730030, China; mengych19@lzu.edu.cn (Y.M.); huangk18@lzu.edu.cn (K.H.); shimx2021@lzu.edu.cn (M.S.); huoyf20@lzu.edu.cn (Y.H.); lhan17@lzu.edu.cn (L.H.); 2Key Laboratory of Preclinical Study for New Drugs of Gansu Province, School of Basic Medical Sciences, Lanzhou University, Lanzhou 730030, China

**Keywords:** tropomyosin, *TPM*, cancer, migration, miRNA, epithelial–mesenchymal transition, proliferation, biomarker, apoptosis

## Abstract

Cancer is one of the most difficult diseases for human beings to overcome. Its development is closely related to a variety of factors, and its specific mechanisms have been a hot research topic in the field of scientific research. The tropomyosin family (Tpm) is a group of proteins closely related to the cytoskeleton and actin, and recent studies have shown that they play an important role in various cancers, participating in a variety of biological activities, including cell proliferation, invasion, and migration, and have been used as biomarkers for various cancers. The purpose of this review is to explore the research progress of the Tpm family in tumorigenesis development, focusing on the molecular pathways associated with them and their relevant activities involved in tumors. PubMed and Web of Science databases were searched for relevant studies on the role of Tpms in tumorigenesis and development and the activities of Tpms involved in tumors. Data from the literature suggest that the Tpm family is involved in tumor cell proliferation and growth, tumor cell invasion and migration, tumor angiogenesis, tumor cell apoptosis, and immune infiltration of the tumor microenvironment, among other correlations. It can be used as a potential biomarker for early diagnosis, follow-up, and therapeutic response of some tumors. The Tpm family is involved in cancer in a close relationship with miRNAs and LncRNAs. Tpms are involved in tumor tissue invasion and migration as a key link. On this basis, *TPM* is frequently used as a biomarker for various cancers. However, the specific molecular mechanism of its involvement in cancer progression has not been explained clearly, which remains an important direction for future research.

## 1. Introduction

The tropomyosin family (Tpm) is a two-chained α-helical coiled-coil actin-binding protein that is widely expressed in muscle and non-muscle cells [1]. Tpm localizes laterally along actin filaments and regulates actin interactions; at the same time, it regulates actin dynamics by binding and modulating the activity of other actin-binding proteins [2]. The role of Tpm in actin is inseparable from Ca^2+^. In skeletal muscles, in the absence of Ca^2+^, Tpm is located above the outer domain of actin, and its position spatially blocks most sites of myosin binding. However, in the presence of Ca^2+^, Tpms move to the internal domain of actin, and Tpm prompts actin to expose most of the previously blocked myosin-binding sites. Based on the continuous influence of actin dynamics in the above ways, spatial segregation and non-redundant functional diversity of the Tpm also regulate and influence the function of the actin cytoskeleton [3,4]. In addition to the direct contribution of actin structures to physical stability and mechanical properties, actin fibers are actively involved in contractile force generation and cell adhesion. Some researchers have found that cancer cells have higher contractility than normal cells [5,6,7]. There is a direct relationship between the invasion of cancer cells and the traction force. The improved traction force, due to the rearrangement of actin structure, production of more stress fibers, and enhanced ATP hydrolysis, allows cancer cells to better penetrate into the extracellular matrix (ECM) through thin, long filamentous processes [8]. The occurrence and progression of cancer are significantly related to cell–cell and cell–ECM adhesion, which provides suitable conditions for the migration, invasion, and proliferation of cancer cells. In addition to this, alterations in actin structure not only enhance the deformability of cancer cells but also the higher traction force and alterations in cell adhesion that are crucial during invasion [9]. Tpms have a role in stabilizing the cytoskeleton and wrapping actin filaments and play a decisive role in the fine motility of almost all actin filament structures [10]. In addition, studies have confirmed that Tpms are also involved in other physiological processes, including cell division, cell motility, apoptosis, and signal transduction [2]. At present, members of the Tpm family have been detected in mammals with four main genetic components, namely, *TPM1*, *TPM2, TPM3*, and *TPM4* [11]. These four *TPM* genes can produce more than 40 Tpm subtypes through selective splicing [12]. Direct differences between *TPM1-4* coding regions and alternate use of variant exons 1, 2, 6, and 9 contribute to the diversity of Tpm isoforms [13]. Products encoded by different Tpm isoforms can be divided into high-molecular-weight (HMW) and low-molecular-weight (LMW) subtypes. The HMW isoform binds to seven consecutive actin subunits, whereas the LMW binds to six actin subunits [14]. Numerous studies have shown that HMW Tpms are expressed at down-regulated levels in cancer, while LMW Tpms are overexpressed in tumor tissues, such as *TPM3* in esophageal squamous cell carcinoma (ESCC), which was shown to be associated with malignant transformation-related invasion and poor survival of malignant cell lines in breast cancer [15]. Altered expression levels of *TPM* genes also promote changes in other genes. Overexpression of Tpm isoforms in undifferentiated B35 adult neuroblastoma cells results in differential expression of a large number of genes. Tpm isoforms modulate gene expression patterns in a subtype-specific manner that is consistent with their ability to control actin filament function [16]. The different isoforms play their unique roles while at the same time being related to each other.

### 1.1. TPM1

The products of the *TPM1* gene (Tpm1.1, Tpm1.3, and Tpm1.4 isoforms earlier designated as α-tropomyosin) are widely expressed actin-binding proteins that are involved in molecular communication on the cell surface and proliferation signaling between normal cells [17]. The *TPM1* gene and its products are strongly associated with cancer, which is usually considered as tumor suppressors, and *TPM1* overexpression induces apoptosis in cancer cells during cancer progression [18]. After bioinformatic and statistical analysis, the differentially expressed genes (DEGs) of ribosomal, RNA, and ubiquitin-related functional pathways resulting from Tpm1.12 overexpression in undifferentiated cells had the highest statistical significance and were also associated with a variety of cancers. Using B35 neuroblastoma as a study subject, changes in *TPM* expression levels can induce changes in the expression levels of a considerable number of genes. For example, overexpression of Tpm1.12 can lead to changes in the expression levels of more than 4000 genes. The regulation of other genes is not similar between different isoforms of Tpm and has a more pronounced homozygous specificity. This feature was more pronounced in differentiated versus undifferentiated cells. In differentiated cells, the number of DEGs was greatly reduced compared to that in undifferentiated cells [16]. For example, in bladder cancer, *TPM1* has a role in inhibiting bladder cancer cell proliferation and promoting apoptosis [19]. It has been reported that *TPM1* overexpression combined with radiotherapy can significantly inhibit the growth of U251 xenografts, suggesting that *TPM1* may be the mechanism of radiation resistance in glioma [20]. In a study on lung cancer, the *TPM1* gene was confirmed to inhibit the proliferation and invasion of lung cancer cells and enhance cell apoptosis through the regulatory effect of circ-RNAs [21]. Similar to these findings, EZH2 is a specific H3K27me3 histone methyltransferase, and EZH2 promotes the expression of H3K27mp3.35. EZH2 has oncogenic effects in human malignancies. *TPM1* is one of the downstream targets of EZH2, and LINC01116 binds and recruits EZH2 to downregulate the expression of *TPM1*, which in turn enhances CRC proliferation and angiogenesis. The mechanism may be that overexpression of LINC01116 in CRC leads to a significant increase in the EZH2-rich *TPM1* promoter and regulates EZH2 in CRC at the post-transcriptional level, which is a transcription factor of *TPM1*. EZH2 in CRC is negatively correlated with *TPM1*, and blocking *TPM1* promotes CRC cell proliferation [22]. It is not the only case; the lncRNA MEG3 regulates both miR-96 and *TPM1*. Its overexpression downregulates miR-96 expression level and upregulates *TPM1* expression, which inhibits cell proliferation and promotes apoptosis in bladder uroepithelial carcinoma cells. When lncRNA MEG3 is lowly expressed, it promotes bladder uroepithelial carcinoma cell proliferation and inhibits apoptosis by co-regulating miR-96 with *TPM1* [19]. *TPM1*, a target gene of miR-96, is also thought to be associated with oxaliplatin resistance in CRC. All indications suggest that miR-96 is a tumor promoter. The miR-96 inhibits the expression level of *TPM1* by targeting its 3’-UTR, based on which CRC cells show significant oxaliplatin resistance. This is another manifestation of *TPM1* as a tumor-suppressive factor [23]. This further supports the tumor-suppressive role of *TPM1*. Notably, regarding the lncRNA-regulatory *TPM1* mechanism, some scholars reported that the alternative splicing mechanism of endogenous *TPM1* exon 2a or 2b was found to be different in ESCC cells than in non-cancerous cells, and they identified a previously unlabeled nuclear lncRNA *TPM1-AS* in human cancer cells (reverse transcribed from the fourth intron region of *TPM1*) and demonstrated that it is involved in alternative splicing and cell motility of *TPM1* mRNA precursor. A novel mechanism of lncRNA involvement in alternative splicing of target genes was proposed by the interaction of *TPM1-AS* lncRNA with RNA-binding motif protein 4 (RBM4). That is, lncRNA interacts with the alternative splicing factor SR to promote the generation of variants [24].

On the other hand, there is a lot of evidence that miRNA and *TPM1* are closely related. miR-558 is upregulated in neuroblastoma, gastric cancer, and other tumors and promotes tumor invasion and peritumor blood supply formation. It targets and inhibits the expression of *TNFAIP1* and *TPM1*, which are considered to be tumor-suppressive factors and are often used as targets of miRNAs including miR-558 and miR-224, thereby allowing tumor progression [21]. The overexpression of miR-21 can promote the invasion and migration of ESCC by inhibiting *TPM1* [25]. Consistent with previous findings, miRNA-21 is overexpressed in renal cell carcinoma (RCC) tissues and regulates the growth, apoptosis, and cell cycle progression of RCC cells, as well as the expression of programmed cell death 4 (PDCD4) and *TPM1* [26]. Further studies have shown that overexpression of *TPM1* in RCC inhibits tumor cell proliferation and promotes tumor cell apoptosis. The reason is that overexpression of *TPM1* may lead to DNA damage in tumor cells, which activates p53 expression, subsequently regulating Bcl-2 family members and finally promoting apoptosis of RCC cells through the mitochondrial pathway [27]. Similar findings were found that miRNA-183-5p.1 promotes the migration and invasion of gastric cancer AGS cells by targeting *TPM1* [28]. miR-107 promotes the survival, migration, and invasion of human osteosarcoma cells by regulating *TPM1* [29] (Figure 1). 

However, it has also been reported that the misexpression of *TPM1* can lead to the fracture of stress fibers, thus changing the morphology and vitality of cells, and eventually leading to the malignant transformation of normal cells [30]. But such results do not seem to affect the widely held belief that *TPM1* plays a positive role in tumor tissues.

### 1.2. TPM2

Among the four Tpm isoforms expressed from the *TPM2* gene, Tpm2.1 and Tpm2.2 were designated as β-tropomyosin in previous works. They are widely expressed in fibroblasts, smooth muscle cells, and skeletal muscle cells which are mainly involved in cell motility and muscle contraction regulation [31]. Research hotspots show that Tpm2.1 is crucial for sensing changes in substrate stiffness [32]. Studies on the abnormal expression levels of *TPM2* in cancer cells are common [33]. It is well known that the occurrence and development of hepatocellular carcinoma (HCC) are closely related to the infection of patients with HBV virus. Studies have found that overexpression of *TPM2* increases the production of HBV, and in HBV-related HCC or acute liver failure, *TPM2* expression increases about 4-fold to over 6-fold, respectively [34,35]. These phenomena suggest that HBV may upregulate *TPM2* expression and regulate the actin cytoskeleton for efficient propagation and/or replication in the liver, thereby promoting the progression of HCC [36].

The study of *TPM2* as a biomarker of tumor tissue is also one of the hot research topics. *TPM2* was identified as one of the fibroblast-specific biomarkers of poor prognosis in CRC [37].

In addition, it is worth noting that *TPM2* also has a particular contribution to promote apoptosis in cancer cells. The exploration of the mechanism has also been added by scholars. Tpm2.1 has been shown to increase the sensitivity of cells to apoptosis by dissociating extracellular matrix (apoptosis) and modulating apoptosis-inducing proteins [38].

### 1.3. TPM3

Among the 10 Tpm isoforms expressed from the *TPM3* gene, only Tpm3.12 (slow skeletal muscle Tpm isoform) was designated as γ-tropomyosin in previous works. Tpm3 mediates the response of myosin to calcium ions and maintains the stability of cytoskeletal microfilaments in cells [39]. There is more and more evidence that overexpression of *TPM3* is strongly associated with cancer occurrence and progression. Similarly, Tpm3.1 has also been shown to be highly specifically upregulated in all cancer cell lines tested to date [10]. For example, Tpm3.1 is abundant in epithelial ovarian cancer tumors of all tissue types [40]. In addition, its combined targeting with microtubules has a strong anti-tumor synergistic effect [41,42]. The potential mechanisms of its aberrant expression in tumor tissues and the resulting effects have been illustrated by a number of studies. Overexpression of Tpm3.1 in undifferentiated cells up-regulated the expression levels of an integrin subunit gene α4 (Itga4) [16]. Fbln5 enhances cell adhesion and reduces proliferation through integrin binding [43,44]. Meanwhile, Fbln5 is reported to have context-dependent oncogenic and tumor-suppressing roles [45,46]. Itga4 can regulate cell migration by forming dimers with b1 subunits and binding to actin IIa [47]. Overexpression of Tpm3.1 may act synergistically with Fbln5, Itga4, and myosin IIa to enhance cell stability and adhesion. 

Part of the reason why *TPM3* is considered an oncogene is that its abnormal expression drives changes in other genes that allow tumor tissues to progress. E-cadherin has long been regarded as a tumor-metastasis-suppressor gene and a key gene in the process of EMT, and its abnormal expression is the molecular basis of cell division [48]. Down-regulation of E-cadherin expression promotes the occurrence of EMT and cell carcinogenesis. Down-regulation of *TPM3* gene expression leads to abnormal activation of E-cadherin and vimentin genes, resulting in morphological changes of pancreatic cancer (PC) cells, increasing the degree of malignancy, and promoting PC metastasis through EMT [49].

In addition, one of the main research directions of *TPM3* is oncogene fusion and gene rearrangement. *TPM3* is frequently involved in gene rearrangements leading to fusion with the neurotrophic tyrosine kinase receptor type 1 (NTRK1) gene, which then acts as an oncogene [50,51]. The “*TPM3-TRK*” fusion oncogene is the result of an intra-chromosomal rearrangement that results in the fusion of the *TPM3* gene with sequences encoding the transmembrane and intracellular domains of the transmembrane tyrosine kinase known as tropomyosin receptor kinase (TRK) [52,53]. Similar studies have reported that anaplastic lymphoma kinase (ALK) is a receptor tyrosine kinase. *TPM3-ALK* and *TPM4-ALK*, a fusion protein consisting of the N-terminal of *TPM* and the C-terminal kinase domain of ALK, have been reported in patients with inflammatory myofibroblastic tumors, with *TPM3-ALK* being the most commonly observed [54].

Targeted therapy for Tpm3.1 has become a hotspot. At present, three active compounds with selective anti-Tpm3.1 activity have been screened: TR100 [55], ATM1001 [56], and ATM350 [41]. Regarding the molecular mechanism of compound ATM-3507 acting on Tpm3.1, it has been shown that 3H-ATM-3507 is incorporated into filaments and saturates at about one molecule per Tpm3.1 dimer with an apparent binding affinity of about 2 µM when it is present during the co-polymerization of Tpm3.1 with actin. Meanwhile, ATM3507 may alter the lateral movement of Tpm3.1 on the actin surface, thereby altering the interaction of filaments with actin-binding proteins and myosin motors [57]. A study on the targeting of compound ATM reported that intervention with compounds (TR100 and ATM1001) in wild-type mice (WT) and Tpm3.1 knockout mice (KO) resulted in reduced glucose clearance (inhibition of glucose-stimulated insulin secretion (GSIS) from pancreatic islets). The results showed that the GSIS of Tpm3.1 knockout mice was significantly less affected than that of wild-type mice, indicating that the drug action was targeted. In cell experiments, it was found that the inhibition of GSIS by the drug was due to the destruction of the cortical actin cytoskeleton. Interestingly, both compounds inhibited insulin-stimulated glucose uptake in WT muscle, but in KO muscle, the drug had little effect [56]. These results indicate that ATM drugs affect glucose metabolism in vivo by inhibiting the function of Tpm3.1, with little off-target effect. This finding may provide new ideas for the research field of inhibiting glucose uptake in cancer cells to delay its development (Figure 2).

In summary, *TPM3* research is relatively common, and its research direction tends to be multi-directional. There are also more and more mechanisms being elucidated by more studies.

### 1.4. TPM4

Among the two Tpm isoforms expressed from the *TPM4* gene, only Tpm4.1 was designated as δ-Tpm in previous works. *TPM4* regulates the contraction of skeletal muscle and smooth muscle cells or maintains the stability of the cytoskeleton in non-muscle cells [50,51]. In tumor tissues, *TPM4* also plays an abnormal function. In undifferentiated rat B35 neuroblastoma, the expression of integrin subunit gene α7(Itga7) was up-regulated with the overexpression of Tpm4.2 [16]. Integrins play a role in cell surface adhesion and signaling and act as a mediator between the actin cytoskeleton and the extracellular matrix [58]. 

In addition, cancers with abnormal *TPM4* expression have been confirmed, including lung cancer [59], breast cancer [60], esophageal cancer [61], ovarian cancer [59], cervical cancer [62], prostate cancer [63], and colon cancer [64].

## 2. Role of Tpm in Tumor Proliferation and Growth

The proliferation and growth of tumors are different from normal cells, but the mechanism is not very clear. More and more research tends to explore this aspect. 

Interestingly, some researchers have found that cancer cells of different tissues can switch between transformed and rigidity-dependent growth states by the presence or absence of mechanosensory modules [65]. Tpm2.1 is involved in the formation of the cell stiffness sensing complex, sarcoma-like contractile units (CUs). The transformation and growth of cancer cells require the consumption of rigid sensing modules. The high level of *TPM3* expression may inhibit CU formation and rigidity perception through competition of its gene products with Tpm2.1 [65]. There is a competitive relationship between Tpm2.1 and gene products of *TPM3*, where *TPM3* overexpression leads to CU deletion. This may be a potential mechanism for *TPM3* to act as an oncogene. It leads to the proliferation and migration of cancer cells by depleting the rigid-sensitive module, which leads to the transformed growth of cancer cells. This suggests that the transformation and growth of cancer cells is a mechanobiological phenomenon. Another study has also provided evidence supporting this notion. In tumor cells, when transformed by the loss of the rigid sensor protein (Tpm2.1), its behavior is similar to that of normal cells. Restoration of rigid sensing in tumor cells promotes rigid-dependent mechanical behavior, that is, cyclic stretching enhances growth on soft surfaces and reduces apoptosis. Thus, tumor cells can be selectively killed by mechanical perturbation while stimulating the growth of normal cells [66].

Actin filaments containing Tpm3.1 mediate nuclear translocation of extracellular signal-regulated kinases, thereby promoting cell growth and proliferation of mouse fibroblasts [67]. During mitosis, Tpm3.1 or Tpm3.1-containing actin filaments are enriched in the cell cortex [42], where stellate microtubules interact with cortical actin through protein complexes (such as NuMA-dynein-dynactin and LGN-gαi) required for mitotic spindle assembly and localization [68]. 

ATM analogs antagonize *TPMs* in tumor tissues with significant targeting properties. The anti-*TPM3* compound ATM-3507 described in the part of the introduction was shown to inhibit tumor growth by targeting the C-terminus of Tpm3.1 [55,57]. Moreover, ATM-3507 does not impair muscle structure and function, suggesting that ATM-3507 may target the cytoskeleton of tumor cells but not muscle tissues [57]. 

Abnormal expression of *TPMs* at the level of *TPMs* is accompanied by aberrant regulation by other cellular regulators as well as mi-RNAs. They have a close association with *TPMs*. The inhibition of miR-107 leads to the elimination of the functional inhibition of its downstream genes, which further promotes the malignant phenotype and chemoresistance of cancer, and is regarded as a tumor-suppressor regulator [69,70]. Current evidence shows that miR-107 is repressed in ESCC tissues and cell lines, whereas *TPM3* is upregulated, especially in advanced ESCC tissues [71]. Dual luciferase reporter analysis showed that miR-107 could directly target *TPM3* mRNA and inhibit *TPM3* protein expression in ESCC cells [72]. These results imply that miR-107 is a negative regulator of ESCC progression by inducing the degradation of *TPM3* mRNA.

Regarding the mechanism of *TPM3* action in ESCC, one line of evidence suggests that PCBP1 is an upstream regulator of *TPM3* [73]. PCBP1 is an RNA-binding protein that is abnormally expressed in a variety of tumor tissues [74,75]. PCBP1 is responsible for the significant upregulation of *TPM3* in esophageal squamous cell carcinoma. PCBP1 is an RNA-binding protein (RBP). PCBP1 is highly expressed in esophageal cancer tissues. Knockdown of PCBP1 significantly attenuates the proliferation, migration, and invasion of ESCC cells by binding directly to the 3′untranslated region (3′utr) of *TPM3* mRNA and stabilizing mRNA degradation by binding directly to position 1317–1322 of *TPM3* mRNA [73].

The specific mechanism of *TPMs* on tumor tissue growth and proliferation is not very completely elucidated. There is more evidence from a number of studies from various sources. Future studies need a main line of sight to connect these compelling pieces of evidence into a complete mechanism.

## 3. Role of Tpm in Tumor Invasion and Migration

It is well known that the invasion and migration of tumor tissues are often the main culprits in the refractory nature of cancer and poor prognosis. From the existing literature, the role played by *TPMs* in tumor invasion and migration cannot be underestimated. They may be involved in a number of physiological processes, and this mystery is slowly being unveiled by scholars. 

The involvement of *TPMs* in this process has been shown in studies related to the mode and mechanism of tumor cell migration, and *TPMs* play a regulated role in this process. Stress fibers not only affect cell morphology and differentiation but also play an important role in the invasion and migration of tumor cells. In osteosarcoma cells, different formin isoforms determine the location of different Tpm isoforms in dorsal stress fibers. DAAM1 and FHOD1 are two subtypes of formins. DAAM1 specifically inhibited the dorsal stress fibers modified by Tpm3.1. Similarly, FHOD1 knockdown significantly changed the localization of Tpm3.1 in the dorsal stress fibers [76]. They do not affect other *TPMs*. This suggests that *TPMs* can be targeted and regulated by formins to change the migration and progression of tumors. 

Likewise, mi-RNAs and *TPMs* have always maintained a close association. As the target genes of mi-RNAs, they are regulated and constantly act to change the state of tumor tissues. Conclusive evidence has shown that miRNA-183 has a high expression level in prostate cancer cells and contributes to the progression of malignant tumors and lymphatic metastasis [77]. Furthermore, miRNA-183 can target the *TPM1* gene and down-regulate its expression, thereby promoting the progression of prostate cancer [78]. This is consistent with previous reports that miR-183-5p promotes tumor metastasis and growth of non-small cell lung cancer (NSCLC) by down-regulating PTEN [79]. There was a negative correlation between the expression of *TPM1* and miR-21 in ESCC. The data indicate that miR-21 targets *TPM1* in ESCC and affects ESCC migration and invasion. miR-21 inhibits *TPM1* expression by binding to the 3 ‘untranslated region (3’ UTR) of *TPM1* mRNA, thus promoting the migration and invasion of ESCC [25]. The underlying mechanism is that *TPM* is a class II tumor-suppressor gene, and its gene sequence structure is complete, but its expression is insufficient or not expressed due to downregulation or silencing in transcription or translation [80].

The expression of miR-107 is up-regulated in osteosarcoma cells U2OS, and its overexpression promotes the survival, migration, and invasion of U2OS cells by down-regulating the *TPM1*-stimulated MEK/ERK and NF-κB signaling pathways [29]. In clinical treatment, Skullcapflavone I was confirmed to affect the cell proliferation of CRC cells by blocking MEK/ERK and NF-κB by regulating *TPM1* and miR-107 [81]. These findings all suggest that the signaling pathways with high relevance to *TPMs* are MEK/ERK and NF-κB.

*SUSD2* is an NSCLC-associated gene, and it is under-expressed in HCC. Interestingly, *SUSD2* abolished the regulatory effect of *TPM4* on HCC cell behavior. Therefore, *SUSD2* is responsible for *TPM4*-induced HCC exacerbation. The overexpression of *TPM4* in HCC tissues aggravates the malignancy of HCC through the negative regulation of SUD-2 [82]. The tumor-suppressive effect of miR-133a may be related to the Mir-133a-dependent regulation of *TPM4* expression. miR-133a → *TPM4* and TAp63γ → *TPM4* axis are key elements in muscle process. The low levels of miR-133a observed in CRC lead to increased *TPM4* expression levels, which are responsible for altered cytoskeletal structure, high cell motility, migration, and metastasis, and may favor EMT [83] (Figure 3).

Another study on the involvement of *TPMs* in tumor tissue invasion and migration also claimed that they were closely associated with EMT. *TPM1* is involved in EMT, a key process in tumor invasion and distant spread. EMT factor can promote tumor cell invasion by upregulating matrix metalloproteinases (MMPs) and downregulating E-cadherin [84]. These results suggest that *TPM1* inhibits RCC cell migration by regulating these molecules [85]. This result also provides evidence for *TPM1* as a tumor-suppressive factor. Down-regulation of MMP2/MMP9 expression can lead to different expressions of EMT markers and inhibit the progression of EMT. During EMT, cancer cells lose their epithelial phenotype and cell polarity and acquire a potent ability to degrade the ECM, resist apoptosis, invasion, and migration, and induce a mesenchymal cell phenotype [86]. *TPM3* can up-regulate the expression of MMP2 and MMP9, induce the progression of EMT, and activate the proliferation, migration, and metastasis of ESCC cells [87].

The essential functions of *TPMs* seem to be reflected in the tumor tissue. A study suggests that the mechanism through which non-muscle *TPMs* interfere with tumor tissue migration in cancer cells may share the same regulatory pathway as myosin. In addition, *TPMs* were found to be closely related to the Rho protein family in the study. Loss of Tpm2.1 has been found to increase Rho-GTP levels and activate Rho-Rock-mediated regulation of actomyosin contractility [88]. Inhibition of Rho-associated kinase (ROCK) reversed the delay in collective cell migration caused by loss of Tpm2.1. In addition, another study found that loss of Tpm2.1 disrupted cell polarity at the leading edge [89]. These results suggest that Tpm2.1 regulates actomyosin contractility through striated muscles and is important for collective cell migration together with cell polarity modulators. Notably, in the study of multicellular invasion, some scholars reported that the expression of Tpm3.1 was negatively correlated with Rac GTPase-mediated cell invasion. Rac GTPases are known intracellular transducers that regulate multiple signaling pathways that control cytoskeletal organization, transcription, and cell proliferation [90]. Combination therapy with Rac inhibition and Tpm3.1 targeting inhibited multicellular invasion more than treatment alone. Data have shown that disruption of Tpm3.1 sensitized neuroblastoma cells to Rac inhibition of multicellular invasion [91]. 

Cyclin Y (CCNY) is a novel cyclin, and there are two subcellular subtypes of CCNY, namely, cytoplasmic subtype (CCNYc) and membrane subtype (CCNYm). CCNYc and CCNYm have also been shown to be expressed at high levels in lung cancer [59]. In addition, CCNY promotes the migration and invasion of various tumor cells, including liver cancer and ovarian cancer [92,93]. Meanwhile, CCNY’s potential CDK partner is PFTK1 [94,95]; in turn, CCNY c/PFTK1 regulates cytoskeletal structure through *TPM4* and promotes cell migration and invasiveness through RoA remodeling in the Rho family of GTPases [59].

There were also significant differences in the expression levels of *TPMs* in tumor tissues with different degrees of metastasis. Its mechanism may be that the abnormal expression levels of related genes lead to protein alterations. Tpm4.1 expression was down-regulated in high metastatic breast cancer cell lines compared with low metastatic cells. *TPM4* expression is reduced in invasive ductal breast cancer compared with ductal breast cancer in situ [60]. To invade the surrounding environment, cancer cells need to separate from the primary tumor, which requires breaking cell-to-cell adhesion. Actin reorganization and cadherin aggregation are important in the formation and maintenance of cell–cell junctions [96]. Loss of Tpm4.1 disrupts cuboidal morphology and actin organization in cell–cell junctions. The molecular mechanism involves the down-regulation of Tpm4.1 which induces Rac1-mediated changes in myosin IIB localization and loss of myosin IIB prevents increased cell migration and disruption of cell junctions after Tpm4.1 silencing [60].

Down-regulation of Tpm2.1 may play a key role in tumor progression by promoting the metastatic potential of tumor cells. In addition, studies not identical to the previous paper on the mechanism of cell growth in rigid substrates have been reported. Glioblastoma (GBM) brain tumors are highly invasive to surrounding healthy brain tissue, so the mechanism of invasion defined using a rigid matrix may not be applicable to GBM transmission. Tpm 2.1 is down-regulated in GBM grown on soft substrates and contributes to GBM colonization in the soft brain environment [97]. In addition, Tpm1.7 was also found to be down-regulated, suggesting that the effect of this isoform is matrix-dependent and that Tpm1.7 induces cell rounding in the 3D collagen gels set for the experiment [97]. These results suggest that Tpm 2.1 deletion in primary patient-derived GBM is associated with prolonged mesenchymal invasion.

It is worth noting that tumor tissues often metastasize through blood, and some scholars have discovered that platelets can be cultured by cancer cells, which are called tumor-cultured platelets (TEPs), and regulate their RNA content or absorb tumor RNA in response to signals from cancer cells, thus leading to changes in transcriptome profile and reflecting pathological progress [98]. In another study, the expression of *TPM3* mRNA was significantly elevated in platelets from breast cancer patients compared with age-matched healthy controls. Interestingly, another study also reported that platelet *TPM3* mRNA was delivered to breast cancer cells via microvesicles and resulted in an enhanced migratory phenotype of breast cancer cells [99]. These results suggest that the upregulation of *TPM3* mRNA in platelets is significantly associated with metastasis in breast cancer patients.

An interesting point is that changes in tissue oxygen content can also indirectly affect the expression level of *TPMs*. *TPM2* expression was down-regulated in breast cancer cells compared with normal breast cells. Hypoxia induces promoter methylation of *TPM2*, which is responsible for its low expression. Hypoxia may regulate cell invasiveness partly through changes in MMP2 expression mediated by *TPM2* downregulation. Importantly, low *TPM2* expression was associated with lymph node metastasis, tumor lymph node metastasis stage, histological grade, and shorter overall survival [100].

The Tpm family is often represented as a target gene or as a member of a signaling pathway in the invasion and metastasis of tumor tissue. Different Tpm isoforms play distinct roles, and the specific molecular mechanisms need to be further investigated.

## 4. Role of Tpm in Tumor Vasculogenesis

It is well known that the progression of tumors is closely related to the blood flow of surrounding tissues. It has been demonstrated that tropomyosin exists on the surface of proliferative-activated endothelial cells during the angiogenic transition and represents a variety of receptors for anti-angiogenic proteins [101]. It belongs to another evolutionarily and structurally relevant protein, high-molecular-weight kininogen (HKa). The histidine–proline-rich glycoprotein (HPRG/HRG) is a single chain (75 kDa) protein that has important functions in regulating immunity, angiogenesis, and coagulation system [102]. It is worth noting that HPRG has a high affinity for tropomyosin [103,104]. In addition to Zn^2+^, the metal complex TetraHPRG (the peptide fragment [Ac-(GHHPH)4-G-NH2] belonging to the H/P domain of HPRG) was formed when the Cu^2+^ level was increased in cancer cells [105,106], and its interaction with HKa enhanced the antiangiogenic properties [107]. 

With the rapid growth of malignant tumors, hypoxic stress appears inside the tumor, which initiates a variety of angiogenic signaling pathways to meet the increasing oxygen demand. *TPM1* was found to be a mediator between 4’-acetamino-4-hydroxyl chalcone and its antitumor effects (including inhibition of angiogenesis) in glioma [108]. Upregulation of *TPM1* inhibits RCC angiogenesis by decreasing vascular endothelial growth factor expression [85].

## 5. Role of Tpm in Tumor Apoptosis

Some researchers have found that overexpression of Tpm2.1 can up-regulate death-related protein kinase 3 (Dapk3) and down-regulate the expression of cell division cycle 37 (cdc37) [16]. These two genes are involved in apoptosis, among which Dapk3 has been shown to exert apoptotic function through the mitochondrial pathway [109] and has tumor-suppressor properties [109,110,111]. cdc37 and heat shock protein 90 can promote the proliferation and survival of cancer cells through synergistic interaction [112]. These findings suggest that Tpm2.1 can regulate the expression levels of these genes, thereby exerting the functions of tumor suppression and apoptosis promotion.

mi-RNA is also involved in this process. miRNA-183 is involved in the post-transcriptional process, is upregulated in gastric cancer, and acts as an oncogene in tumor migration. miR-183-5p.1 targets the 3′UTR of *TPM1*. Overexpression of miR-183-5p.1 promotes cell proliferation, migration, and invasion by down-regulating *TPM1* and activating Bcl-2/P53 signaling pathway in gastric cancer. Further analysis of apoptotic signaling proteins showed that Bcl-2 and P53 were involved in the inhibitory effect of miR-183-5p.1 and *TPM1* on AGS cells. These results suggest that the inhibitory effect of miR-183-5p.1 on the expression of *TPM1* may be related to the activation of the apoptotic signaling pathway [28]. However, miR-183 expression was significantly reduced in GC cells and inhibited GC invasion, suggesting that although they belong to the same miRNA-183 family, it may be due to differences in targeting sites resulting in opposite effects on cancer cells. On the other hand, LncRNA-MEG3 is a miR-96 sponge in bladder cancer that adsorbs miR-96 expression and upregulates *TPM1* expression, inhibits cell proliferation, delays the cell cycle, and promotes apoptosis. miR-96 directly targets *TPM1* and downregulates its expression. This formed a regulatory network: lncRNA MEG3 sponge adsorbed miR-96 and directly increased the expression of tumor-suppressor *TPM1* [19].

There are several similar studies. Tropomyosin subtype Tpm2.1 is an important regulator of cell exit apoptosis (anoikis) [32,33]. Caspase 3/7 activation and mitochondrial depolarization were specifically increased, indicating that overexpression of Tpm2.1 enhances sensitivity to apoptosis by regulating apoptotic signaling rather than by changes in actin filament dynamics [38]. Tpm2.1 loss in cancer cells may be part of the apoptosis evasion mechanism. In this study, Tpm2.1 was not only confirmed to be a sensor of anoikis but was also shown to enhance sensitivity to intrinsic apoptosis by enhancing the degradation of the pro-survival proteins, Mcl-1 and Bcl-2 [38].

## 6. The Relationship between Tpm and Immunity in Tumor Microenvironment

It has been documented that in HCC, due to up-regulation of *TPM*, B and T cells may be recruited to the tumor site. Among them, *TPM3* is negatively correlated with M2 macrophages [113]. M2-like macrophages can promote the proliferation and invasion of HCC cells by activating TLR4/STAT3 signaling pathway [114]. This implies that *TPMs* may exert inhibitory effects on HCC cells by promoting the polarization of M2 macrophages.

The expression levels of *TPM1*, *TPM2*, and *TPM4* genes were decreased in bladder cancer cells. However, they were in direct proportion to immune cells, including NK cells, macrophages, neutrophils, and Th1 cells. This may contribute to immune suppression in bladder cancer [115].

Through bioinformatics analysis, *TPM2*, as one of the six immune-related genes, is significantly associated with the prognosis of colon cancer and constitutes an immune-related prognosis model of colon cancer, which plays a key role in the tumor immune microenvironment [116].

In PC, high levels of *TPM4* expression have been found to be strongly associated with poorer overall survival, disease-specific survival, disease-free survival, and progression-free survival [117]. These findings strongly suggest that *TPM4* may be an oncogene and prognostic biomarker for PC. Some studies have found that pancreatic cancer patients with DNA mismatch repair defects (MMRDs) have specific clinical, pathological, and genomic features [118]. The expression of *TPM4* was also confirmed to be significantly correlated with the mutation levels of five MMR genes in pancreatic cancer and the level of immune infiltration in the tumor microenvironment [117].

## 7. *TPM* as a Tumor Biomarker

As we all know, tumor biomarkers have a role that cannot be ignored in clinical diagnosis and have an important contribution to clinical therapy. Some scholars found that *TPM1, TPM2, TPM3*, and *TPM4* genes are expressed in liver cancer tissues with an up-regulated expression level. The risk model showed that *TPM1, TPM2*, and *TPM3* were used to assess the prognosis of HCC, and *TPM1* was negatively correlated, which means that *TPM1* may be a good prognostic factor for HCC. The high expression of *TPM3* is associated with poor survival outcomes in patients with liver cancer [113]. However, another study showed that *TPM1* expression was down-regulated in intrahepatic cholangiocarcinoma (ICC) compared with adjacent normal tissues. At the same time, *TPM1* is also associated with the TNM stage of tumors. Patients with high *TPM1* expression have a better survival rate. After univariate and multivariate analysis, *TPM1* could be considered as an independent prognostic factor of ICC [119]. Similarly, *TPM3* and *TPM4* expression levels are also up-regulated in gliomas and are closely associated with poor prognosis. Among them, *TPM3* can be used as a new independent prognostic factor for glioma [120].

It is worth noting that although *TPM1* and *TPM2* are highly expressed in normal urothelial tissues, the expression of *TPM1* and *TPM2* is decreased in the early stage of bladder cancer, which may be a marker event for the occurrence and development of bladder cancer [121]. Meanwhile, the expression levels of *TPMs* in low-grade bladder cancer were lower than those in high-grade bladder cancer [115]. Therefore, *TPM1* and *TPM2* are effective markers for the diagnosis of bladder cancer and are expected to be potential therapeutic targets for bladder cancer.

There is solid evidence that *TPM1* can act as a tumor-suppressor gene in oral squamous cell carcinoma (OSCC). The expression level of *TPM1* in adjacent normal tissues was significantly higher than that in OSCC lesions. At the same time, high expression of *TPM1* can slightly inhibit cell proliferation, strongly inhibit cell mobility, and significantly promote cell apoptosis. This implies that down-regulated *TPM1* is an important marker of poor prognosis in OSCC. It is worth noting that miR-21 has been studied in solid tumors and upregulated in tongue cancer [122,123], but luciferase detection did not reveal results that could be used to verify the association between miR-21 and *TPM1*. It may be that miR-21 regulates the expression of *TPM1* while promoting OSCC [124]. Further studies are needed to elucidate the underlying molecular mechanisms.

Human tumor-associated stromal cells (TASCs) isolated from CRC tissues trigger EMT of tumor cells in vitro and promote metastasis and spread in vivo [125]. Expression of calponin 1 (*CNN1*) and *TPM2* was significantly associated with adverse outcomes in independent databases. Recent studies based on single-cell gene expression analysis in colorectal cancer surgical specimens have consistently identified both *CNN1* and *TPM2* as TASC markers associated with poor prognosis [37,126]. It has been demonstrated that TASC overexpressing *CNN1* and *TPM2* at the protein level indeed directly supports cancer progression by enhancing the proliferation, migration, and metastatic potential of tumor cells [125].

In PC, high levels of *TPM4* expression have been found to be strongly associated with poorer overall survival, disease-specific survival, disease-free survival, and progression-free survival. These findings strongly suggest that *TPM4* may be an oncogene and prognostic biomarker for PC.

As we all know, alpha-fetoprotein (AFP) is very important for the diagnosis of liver cancer. However, for AFP-negative HCC patients, some scholars have proposed P53, MSH2, and the product of the *TPM4* gene and inflammatory markers involved by *TPM4* as diagnostic models [127]. Among them, P53 has high specificity for the diagnosis of primary liver cancer [128,129]. MSH2 is hardly expressed in normal hepatocytes, but its expression in HCC gradually increases with the progression of HCC [127]. The increase in the product of the *TPM4* gene is mainly associated with an increase in ferritin source or clearance barrier [127]. Compared with the normal-liver-immortalized human urothelial cell line, the content of the product of the *TPM4* gene in the three HCC cell lines was significantly up-regulated [130].

In a study of colon cancer, *TPM4* expression was shown to be reduced in colon cancer tissues and cell lines. Low expression of *TPM4* was significantly correlated with clinical stage, depth of invasion, lymph node metastasis, distant metastasis, and differentiation. In addition, up-regulation of *TPM4* expression can inhibit the expression of genes related to migration, invasion, and metastasis of colon cancer cells [131]. Compared with normal colon tissues and colonic epithelial cell lines, the mRNA and protein expression of *TPM4* in colon cancer tissues were decreased, respectively [64].

Tpm1.6 and Tpm1.7 proteins were significantly down-regulated in ESCC tumor tissues, but their expression levels were higher in adjacent normal tissues [132]. This is consistent with the downregulation of HMW Tpms reported in breast [133], bladder [115], colon [134], neuroblastoma [135], and prostate cancer [136] studies. However, it is worth noting that an interesting report is that a confounding concept of cellular identity (CCI) has been proposed for ESCC. The average number of genes detected in malignant cells was significantly higher than that in normal cells, and this new cellular identity relative to normal cells was termed CCI and used as an independent marker associated with poor prognosis in ESCC. On this basis, *TPM4* was found to generate CCI by activating the Jak/STAT-SOX2 pathway, and thus, *TPM4* was identified as a key CCI gene promoting ESCC invasiveness [137].

From the existing studies, *TPMs* have slowly become a research hotspot as biomarkers of tumors, which indicates that *TPMs* have significant clinical significance in the early diagnosis and prognosis of tumors. This also suggests that scholars in this field should shift more attention to *TPMs*.

## 8. Methods and Materials

PubMed and Web of Science were used to search for articles related to Tpm and cancer in the past 5 years and 694 articles were obtained. After screening, 564 articles were not related to the content of this study. Finally, 130 articles were obtained and included in the research scope, including 14 articles on Tpm and tumor proliferation and growth; 28 articles on Tpm and tumor invasion and migration; 9 articles on Tpm and tumor angiogenesis; 6 articles on Tpm and immune cell infiltration in tumor microenvironment; 10 articles on Tpm and tumor apoptosis; and 22 articles on TPM as a tumor biomarker. (There are a number of articles covering multiple aspects.)

## 9. Conclusions

In conclusion, Tpm is more involved in the invasion and migration of tumor tissues and has been studied mainly in tumor tissues, such as esophageal cancer and liver cancer. Meanwhile, the activities of Tpm in cancer are closely related to miRNAs and LncRNAs and are associated with epithelial–mesenchymal transition and apoptosis, etc. *TPM* is more often regulated as a downstream gene of oncogenes and can be regarded as a “dependent variable” of cancer development, while *TPM3* is more specific and often appears as an oncogene. *TPM3* is distinctive and often appears as an oncogene from the perspective of researchers. In general, the specific mechanism of Tpm in cancer is not clear, which will be an important direction for future Tpm research.

## Figures and Tables

**Figure 1 ijms-24-13295-f001:**
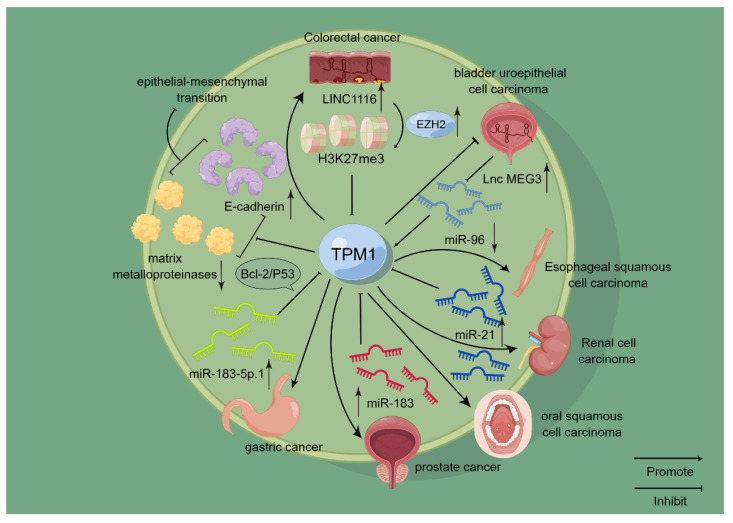
Schematic representation of the mode of action of *TPM1* in selected cancers. *TPM1* has been confirmed to play a role in a variety of cancers, including kidney cancer, oral cancer, bladder cancer, and gastric cancer. Among them, it is closely related to the miRNA family. For example, *TPM1* is negatively correlated with miR-21 and miR-183, and it is positively correlated with miR-96, thereby affecting the occurrence and development of related cancers. On this basis, *TPM1* is also closely related to EMT and acts as a tumor suppressor to some extent.

**Figure 2 ijms-24-13295-f002:**
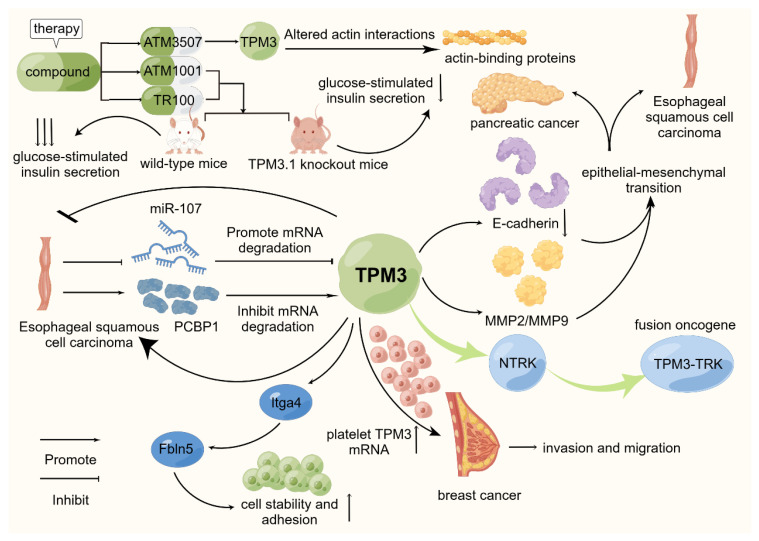
Schematic representation of the mode of action of *TPM3* in selected cancers. *TPM3* is widely recognized as an oncogenic factor in existing studies. It is actively involved in promoting the proliferation, invasion, migration, and other biological behaviors of various cancers. On this basis, *TPM3-TRK*, as a fusion oncogene, is currently a hot topic in related research fields. Through existing studies, *TPM3* has been confirmed to be involved in the occurrence and development of esophageal squamous cell carcinoma, breast cancer, and other tumors. At the same time, some scholars have studied related targeted drugs and achieved good results.

**Figure 3 ijms-24-13295-f003:**
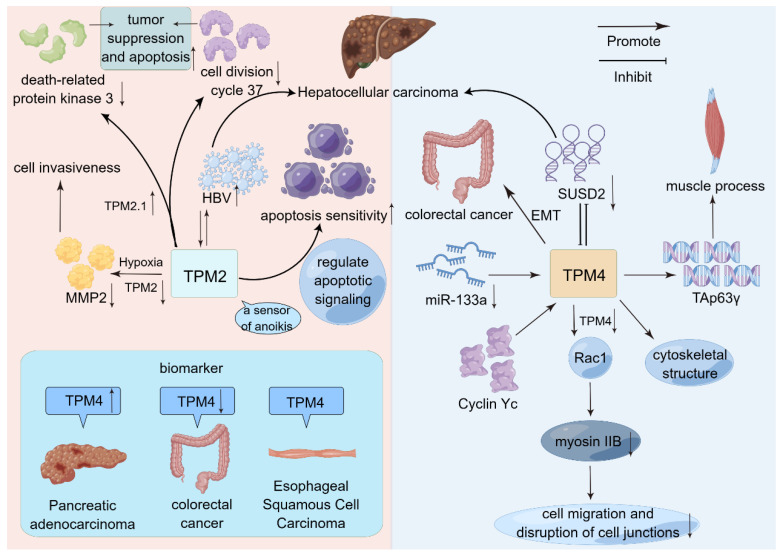
Schematic representation of the mode of action of *TPM2* and *TPM4* in selected cancers. Both *TPM2* and *TPM4* promote HCC. However, *TPM4* promotes HCC progression by interacting with *SUSD2*. *TPM2* promotes HCC progression by interacting with HBV. On this basis, *TPM2* is closely related to apoptosis. In addition to a series of biological behaviors, *TPM4* has more appeared in the public eye as a tumor biomarker.

## Data Availability

Not applicable.

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
