# Peer review of "Research Advances in the Role of the Tropomyosin Family in Cancer"

_ijms, 2023, doi:10.3390/ijms241713295_

Round 1

Reviewer 1 Report

The review-article of Meng et al. summarizes and carefully analyzes a huge and growing body of literature data on the participance of various tropomyosin (Tpm) isoforms in the development of cancer tumors. Generally, this review-article is comprehensive as it covers different aspects in the field of Tpm role in various cellular processes associated with cancer development, such as tumor proliferation and growth, tumor invasion and migration, and tumor vasculogenesis. The literature data on the role of Tpm in tumor apoptosis are also analyzed, as well as the relationship between Tpm and immunity in tumor microenvironment and the role of Tpm as possible tumor biomarker. The paper can be divided into two main parts. In the first part (section 1. Introduction) the authors briefly analyzed the properties of various Tpm isoforms, the products of four genes (TPM1, TPM2, TPM3, and TPM4), and the role of these isoforms in the development of cancer tumors. In the second part (sections 2–7) the authors considered various cellular processes associated with cancer development and the Tpm role in these processes.

The authors have found, using PubMed and Web of Science databases, 89 scientific papers directly related to the main items of the present review-article, and analyzed them in detail. The manuscript is good-organized and illustrated by 3 good-quality figures.

In my opinion, this review-article, which summarizes and carefully analyzes recent literature data on the Tpm role in the development of cancer tumors, will be useful for many scientists working in the field of Tpm functions in the cells, and therefore it can be published in IJMS (particularly, in the special issue "Genes and Human Diseases", section: "Molecular Genetics and Genomics") after some corrections according to the following remarks.

In many cases, it is difficult to understand what the authors mean when write "TPM", the genes (TPM1, TPM2, TPM3, or TPM4) expressing numerous Tpm isoforms or the proteins (Tpm isoforms) expressed from these genes. According to commonly used systematic nomenclature for mammalian Tpm isoforms (Geeves et al. 2015, J. Muscle Res. Cell Motil. 36:147-153, Ref. 43, see also Ref. 44: Gunning et al. 2015, J.Cell Sci. 128:2965-2974), the Tpm (but not TPM!) isoforms expressed from these genes are designated as Tpm1.1, Tpm2.2, Tpm3.12, Tpm4.2, etc.

The authors have should carefully check and correct the names of genes and Tpm isoforms in the manuscript to avoid confusion. See below some examples:

 1). Page 2, section 1.1.: TPM1: "TPM1 (α-tropomosin) is a widely expressed actin-binding protein....". Among 13 Tpm isoforms espressed from the TPM1 gene, only Tpm1.1, Tpm1.3, and Tpm1.4 (skeletal, cardiac, and smooth Tpms) were designated in previous works as α-Tpm. Therefore it would be better to write like that: "The products of the TPM1 gene (Tpm1.1, Tpm1.3, and Tpm1.4 isoforms earlier designated as α-Tpm) are widely expressed actin-binding proteins....".

 2). Page 4, section 1.2. TPM2: "TPM2 (β-tropomyosin) is a protein that is widely expressed in fibroblasts, smooth muscle cells, and skeletal muscle cells...". - The same as in (1). Among 4 Tpm isoforms espressed from the TPM2 gene, only Tpm2.1 and Tpm2.2  were designated in previous works as β-Tpm.

 3). Page 4, section 1.3. TPM3: "TPM3 (γ-tropomyosin).....". - Again, the same as in (1 and 2). Among 10 Tpm isoforms espressed from the TPM3 gene, only Tpm3.12 (slow skeletal muscle Tpm isoform) was designated in previous works as γ-Tpm.

 4). Page 6, section 1.4. TPM4 (should be written in italic!):  " TPM4 (δ-tropomyosin).....". - Again, the same as in (1-3). Among 2 Tpm isoforms espressed from the TPM4 gene, only Tpm4.1 was designated in previous works as δ-Tpm.

5). Page 2: "TPM1 is strongly associated with cancer.....". Who is associated - TPM1 gene or some its products?

"...TPM1.12 overexpression...", "...overexpression of TPM1.12....

It should be the Tpm1.12 isoform, but not TPM1.12!

"...TPM1 was confirmed to inhibit the proliferation..".

Who inhibits: some product/s of the TPM1 gene or the gene itself?

 6). Page 4: "TPM3.1 has also been shown to be highly specifically upregulated in all cancer cell lines tested to date...";  Overexpression of TPM3.1..."

It should be the Tpm3.1 isoform, but not TPM3.1!

 7). Page 5:  "TPM3.1 has become a hotspot...."; "TPM3.1 knockout mice..."

It should be the Tpm3.1 isoform, but not TPM3.1!

 8). Page 6:  "overexpression of TPM4.2..."

It should be the Tpm4.2 isoform, but not TPM4.2!

9). Page 6, section 2: "Role of TPM in tumor proliferation and growth"

It seems much better here, above and below to write tropomyosin (Tpm) as "Tpm", but not "TPM", to avoid some confusion between the protein (Tpm) and the gene (TPM1, TPM2, TPM3, or TPM4) from which it is expressed.

10). Page 6: "TPM2.1 is involved..."; "...rigid sensor protein (TPM2.1)..."; ..."TPM3.1 or Tpm3.1-containing actin filament..."; "...C-terminus of TPM3.1...".

In all these case, it should be the Tpm2.1 and Tpm3.1 isoforms, but not TPM2.1 and TPM3.1!

 and so on in all the text....

Thus, the gene should be TPM (TPM1, TPM2, TPM3, or TPM4), and the protein should be Tpm (Tpm1.1, etc.).

 If you don't know a concrete Tpm isoform produced from the gene (e.g. in some papers published before 2015), it would be better to write like that: ".. the product of the TPM1 (or TPM2, TPM3, TPM4) gene".

11). Page 10, section 5: "...overexpression of TPM2.1"; "TPM2.1 can regulate..."

It should be the Tpm2.1 isoform, but not TPM2.1!

 12). Page 11, section 6: "The expression levels of TPM1, TPM2, and TPM4 were decreased..."

Maybe, it would be better to indicate that these are the genes, as follows: "The expression levels of TPM1, TPM2, and TPM4 genes were decreased..."

 13). Page 12, section 7: "Tm-4 protein molecules..."; "The increase of TPM4...."; "...the content of TPM4 in the three HCC cell lines..."

The paper you cited (Ref. 120) was published in 2013, i.e. before 2015, and therefore it used old nomenclature of Tpm isoforms. To date, only two Tpm isoforms espressed from the TPM4 gene are known, Tpm4.1 and Tpm4.2. If you do not know which of them was described in the citing paper, it would be better to write something like that: "...Tpm isoform, the product of the TPM4 gene."

Minor remarks

 1). Page 2, first line of 1.1. TPM1 section: to replace "TPM1 (α-tropomosin)" by "TPM1 (α-tropomyosin)"

 2). Page 10, section 4: "Zn2+", "Cu2+". These should be Zn2+, "Cu2+" (2+ should be superscript).

 3). Page 13, section Methods and Materials:  "28 literatures", "10 literatures", etc.

It seems better to use "papers" or "articles" instead of "literatures". 

Reviewer 2 Report

In this review paper, the authors described the roles of the tropomyosin family (TPM), which is a group of actin-associated proteins, in cancer cell behaviors. The manuscript was well written and the important publications including the latest ones were cited. However, I have several comments, which would be required for publication.

(1)  The authors should explain in more detail the functions of TPMs in the actin cytoskeleton. Relatedly, the roles of actin in cancer behaviors also need to be explained.

(2)  It is difficult for me to realize what the authors want to show in Figure 2. Illustrations of the mice should not be shown in the figure which indicates TPM3 functions.

I found the several mistakes in the manuscript. In addition, there were some "it" that I was not sure what the author wanted to refer to.

Please carefully amend in the revised manuscript.

Round 2

Reviewer 1 Report

The authors have carefully check the manuscript and corrected the text according to comments done for the previous version. In my opinion, the paper can be published in IJMS (particularly, in the special issue "Genes and Human Diseases", section: "Molecular Genetics and Genomics") in the present form.